# Implications of successive blood feeding on *Wolbachia*-mediated dengue virus inhibition in *Aedes aegypti* mosquitoes

Rebecca M. Johnson [1,5] ✉, Mallery I. Breban [2,5], Braiya L. Nolan[2], Afeez Sodeinde[2], Isabel M. Ott [2], Perran A. Ross [3], Xinyue Gu [3], Nathan D. Grubaugh [2], T. Alex Perkins [4,6], Doug E. Brackney [1,2,6] ✉ & Chantal B. F. Vogels [2,6] ✉

*Wolbachia* is a promising strategy to inhibit dengue virus (DENV) transmission by *Ae. aegypti* mosquitoes. Laboratory studies assessing DENV inhibition by *Wolbachia* typically have not considered natural frequent mosquito blood feeding behavior. Here, we determine the impact of successive feeding on DENV-2 transmission by *Ae. aegypti* in the presence or absence of *Wolbachia* (*w*AlbB and *w*MelM strains). We show that successive feeding shortens the extrinsic incubation period (EIP) in wildtype (WT; without *Wolbachia*) and *w*AlbB mosquitoes through enhanced dissemination. Feeding empirical data into models showed that successive feeding increases the probability of WT and *w*AlbB mosquitoes surviving beyond the EIP. Importantly, the more epidemiologically relevant comparison of the odds of *w*AlbB mosquitoes surviving beyond the EIP relative to WT, reveals a larger impact of successive feeding on WT than *w*AlbB. This indicates a strong inhibitory effect of *Wolbachia* even in the context of natural frequent mosquito blood feeding behavior.

Dengue virus (DENV) is a mosquito-borne virus that poses a significant public health threat, which is exemplified by the current outbreak and a record number of reported cases during the first half of 2024 alone[1,2]. To mitigate this increasing burden, novel control strategies are needed to disrupt DENV transmission between *Ae. aegypti* mosquitoes and humans. One such control strategy is the release of *Ae. aegypti* mosquitoes transinfected with different strains of the virus-inhibiting *Wolbachia pipientis* bacterium[3,4]. *Ae. aegypti* populations transinfected with different variants of the *Wolbachia* *w*Mel or *w*AlbB strains are currently released in the field to either suppress or replace local mosquito populations[5-8]. Mosquito population replacement strategies rely on the ability of *Wolbachia* to inhibit mosquito-borne virus

transmission by mosquitoes, including DENV. However, virus inhibition can be incomplete, and the underlying mechanisms are not fully understood, which warrants further investigation into the underlying factors that may influence the efficiency of virus inhibition[9].

Recently, we found that natural mosquito feeding behavior can influence virus dissemination (exit from the mosquito midgut to secondary tissues such as the salivary glands) and the extrinsic incubation period (EIP; the duration between virus acquisition and transmission)[10-13]. If mosquitoes feed frequently, as is often seen in *Ae. aegypti* in the wild, virus disseminates from the mosquito midgut faster, resulting in a longer time period in which mosquitoes can transmit virus to susceptible hosts, and a larger projected $R_0$[10,11,14-16].

[1]Department of Entomology, The Connecticut Agricultural Experiment Station, New Haven, CT, USA. [2]Department of Epidemiology of Microbial Diseases, Yale School of Public Health, New Haven, CT, USA. [3]Pest and Environmental Adaptation Research Group, Bio21 Institute and the School of BioSciences, The University of Melbourne, Parkville, VIC, Australia. [4]Department of Biological Sciences, University of Notre Dame, Notre Dame, IN, USA. [5]These authors contributed equally: Rebecca M. Johnson, Mallery I. Breban. [6]These authors jointly supervised this work: T. Alex Perkins, Doug E. Brackney, Chantal B. F. Vogels. ✉e-mail: rebecca.johnson@ct.gov; doug.brackney@ct.gov; chantal.vogels@yale.edu

Although some strains of *Wolbachia* can be highly effective in disrupting the transmission of DENV and other mosquito-borne viruses, many of these experiments did not offer mosquitoes more than one blood meal before assessing transmission ability, and current modeling of *Wolbachia* efficacy frequently relies on dissemination times calculated from mosquitoes fed a single blood meal[4,17,18]. While previous studies with *Wolbachia*-transinfected mosquitoes have either provided successive non-infectious blood meals followed by an infectious feed, or provided successive infectious blood meals following an extended egg quiescence, no studies have determined the impact of successive feeding after an initial infectious blood meal[19,20].

In this study, we investigated the effect of successive blood feeding on the inhibition of DENV-2 in *Ae. aegypti* mosquitoes stably transinfected with *w*MelM and *w*AlbB *Wolbachia* strains. Specifically, we evaluated the hypothesis that successive blood feeding decreases the effectiveness of *Wolbachia* by facilitating more efficient DENV

dissemination. Our work has implications for understanding *Wolbachia*-mediated virus inhibition, the use of *Wolbachia*-transinfected mosquitoes for population replacement, and further DENV control efforts.

## Results

### Successive feeding increases DENV-2 dissemination

To test our hypothesis that successive feeding results in more efficient virus dissemination, we provided a DENV-2 spiked infectious blood meal to wildtype *Ae. aegypti* mosquitoes lacking *Wolbachia* (WT), and *Aedes aegypti* mosquitoes stably transinfected with *w*MelM (*w*MelM) or *w*AlbB (*w*AlbB) followed by a second non-infectious blood meal in the "double-feed" group (Fig. 1a). First, we compared infection and dissemination rates across colonies and feeding groups to establish *Wolbachia* inhibition phenotypes and the impact of successive blood meals. In both single-fed and double-fed groups, we

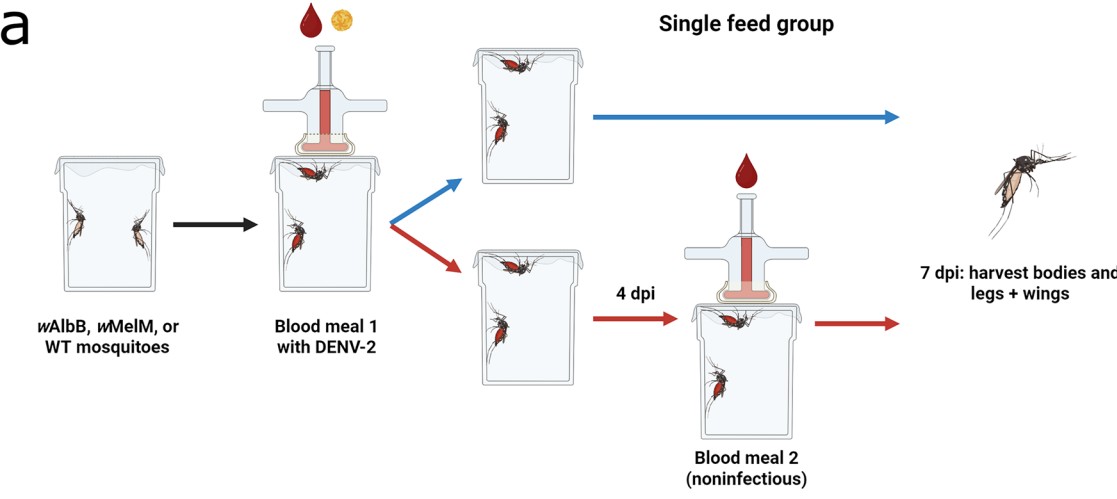

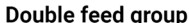

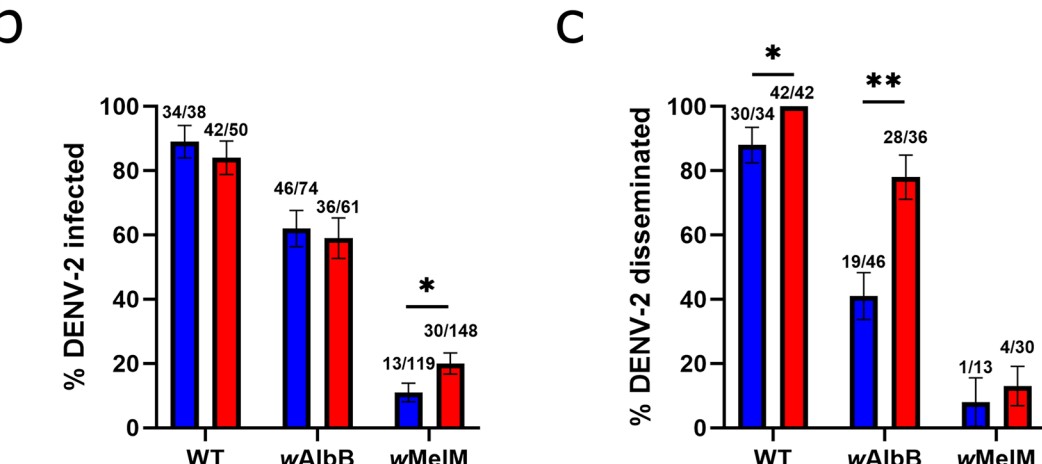

**Fig. 1 | Transinfection with *w*AlbB or *w*MelM *Wolbachia* reduces DENV-2 infection in *Ae. aegypti* and successive feeding leads to higher rates of dissemination in wildtype (WT) and *w*AlbB mosquitoes at 7 dpi. a** Experimental design for initial infection and dissemination studies of single- and double-fed wildtype mosquitoes lacking *Wolbachia* (WT), *w*AlbB, and *w*MelM mosquitoes. *Created in BioRender. Brackney, D. (2025)* https://BioRender.com/bg8atjz. **b** Proportion of infected single- and double-fed WT, *w*AlbB, and *w*MelM mosquitoes 7 dpi. Numbers indicate infected mosquitoes over total fed mosquitoes. *w*MelM single-fed vs double-fed p = 0.0448. **c** Proportion of single- and double-fed

WT, *w*AlbB, and *w*MelM mosquitoes with disseminated infection 7 dpi. Numbers indicate mosquitoes with disseminated infection (measured using legs + wings) over infected mosquitoes. WT single-fed vs double-fed p = 0.0361 and *w*AlbB single-fed vs double-fed p = 0.0015. Comparisons were made using two-sided Fisher's exact tests. *p ≤ 0.05, **p ≤ 0.01, ***p ≤ 0.001, ****p < 0.0001. Blue = single-fed, red = double-fed. Lines indicate mean ± standard error of the mean of the total sample proportions. Data was collected across 4 replicates for WT and *w*AlbB groups and 5 replicates for *w*MelM groups. Source data for (**b**, **c**) are provided as a Source Data file.

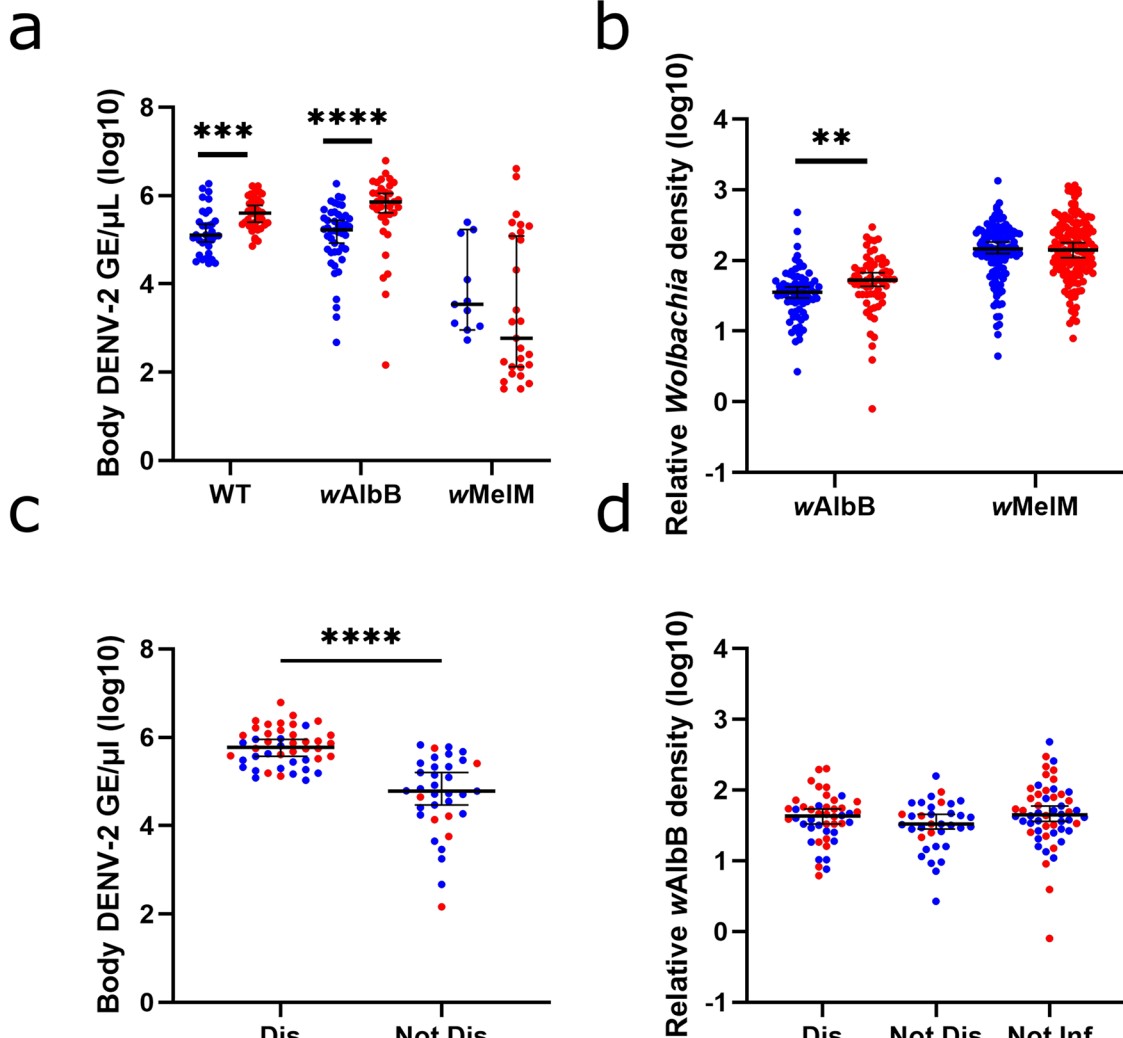

**Fig. 2 | Comparison of DENV-2 titers and relative *Wolbachia* densities at 7 dpi by feeding and dissemination status. a** Body DENV-2 titers in single- and double-fed wildtype mosquitoes lacking *Wolbachia* (WT), *w*AlbB, and *w*MelM mosquitoes. WT SF n = 34, WT DF n = 42, *w*AlbB SF n = 46, *w*AlbB DF n = 35, *w*MelM SF n = 11, *w*MelM DF n = 27. WT single-fed vs double-fed p = 0.0001 and *w*AlbB single-fed vs double-fed p < 0.0001. **b** Relative body *Wolbachia* density in single- and double-fed *w*AlbB and *w*MelM mosquitoes. Numbers tested are as follows: *w*AlbB SF n = 74, *w*AlbB DF n = 61, *w*MelM SF n = 119, and *w*MelM DF n = 148. *w*AlbB single-fed vs double-fed p = 0.0027. **c** Body DENV-2 titers in *w*AlbB mosquitoes by dissemination status and feeding status. Total Dis n = 46 and Not Dis n = 35. Dis SF n = 18, Dis DF n = 28, Not Dis SF n = 28, and Not Dis DF n = 7. Dis vs Not Dis p < 0.0001. **d** Relative body *Wolbachia* densities in *w*AlbB mosquitoes by infection, dissemination, and feeding status. Total Dis n = 46, Not Dis n = 35, and Not Inf n = 54. Dis SF n = 18, Dis DF n = 28, Not Dis SF n = 28, Not Dis DF n = 7, Not Inf SF n = 28, and Not Inf DF n = 26. Comparisons were made using two-tailed Mann-Whitney U tests (**a**–**c**) or a Kruskal-Wallis test with Dunn's multiple comparisons (**d**). *p ≤ 0.05, **p ≤ 0.01, ***p ≤ 0.001, ****p < 0.0001. Blue = single-fed, red = double-fed. GE = genome equivalents, Dis = mosquitoes with a disseminated infection (i.e., dengue infected bodies and legs), Not Dis = mosquitoes with a non-disseminated infection (i.e., dengue infected bodies, but not legs), Not Inf = mosquitoes that were exposed but are not infected with DENV-2. Lines indicate median with 95% confidence interval. Data was collected across 4 replicates for WT and *w*AlbB groups and 5 replicates for *w*MelM groups. Source data are provided as a Source Data file.

found intermediate DENV-2 inhibition in *w*AlbB mosquitoes and stronger inhibition in *w*MelM mosquitoes as compared to those without *Wolbachia* (Fig. 1b, c). This is consistent with dengue inhibition phenotypes previously reported for these *Wolbachia* strains[3,21]. For WT mosquitoes, we observed no difference in infection rates between the single- and double-fed groups, whereas dissemination was increased in the double-fed group, consistent with our previous work (Fig. 1c)[10]. For *w*AlbB, we found a similar effect with a significant increase in dissemination after the second blood meal (Fig. 1c). While we found similar trends for *w*MelM, strong inhibition resulted in low numbers of infected mosquitoes and even lower numbers with subsequent dissemination, and therefore we were unable to detect any significant differences between the *w*MelM single- and double-fed groups. Our findings show that successive

blood feeding results in increased dissemination at 7 days post-infection (dpi) in both the presence and absence of *w*AlbB *Wolbachia*.

## Successive feeding increases DENV-2 titer and *w*AlbB density

To explore the basis of increased dissemination after successive blood feeding, we measured DENV-2 titers and *Wolbachia* densities across treatment groups. For WT and *w*AlbB groups, taking a second, non-infectious blood meal led to higher DENV-2 genome equivalents/μL as measured via qPCR, whereas we observed no significant differences in DENV-2 levels between single- and double-fed mosquitoes transinfected with *w*MelM (Fig. 2a). As previous studies have suggested that *Wolbachia* density can impact *Wolbachia*-based virus inhibition, we also examined

relative *Wolbachia* density as calculated using previously described methods in single- and double-fed *w*AlbB and *w*MelM mosquitoes[22,23]. *Wolbachia* density was slightly higher in double-fed *w*AlbB mosquitoes than in single-fed mosquitoes, whereas no difference was detected in *Wolbachia* density between single- and double-fed *w*MelM mosquitoes (Fig. 2b). When individual mosquitoes were examined, there was no correlation between DENV-2 and *Wolbachia* levels for single- and double-fed *w*AlbB mosquitoes (Fig. S1a, b). This trend also held true for single- and double-fed *w*MelM mosquitoes (Fig. S1c, d). This lack of a strong link between DENV-2 levels and *Wolbachia* density led us to more closely investigate whether the differences in DENV-2 levels between single- and double-fed mosquitoes were due to dissemination status or *Wolbachia* density. Given that we measured DENV-2 infection by testing the mosquito carcass, absent the legs and wings, DENV-2 particles that disseminated to other susceptible tissues outside of the midgut would also be included in these measurements. Investigating further, we found that when *w*AlbB mosquitoes were sorted by dissemination status, mosquitoes with a disseminated infection were found to have a higher level of DENV-2 than those with a non-disseminated infection (Fig. 2c). Additionally, there were no differences in *Wolbachia* densities between *w*AlbB mosquitoes that were not infected, or with either non-disseminated or disseminated infections (Fig. 2d). Though numbers were limited, we saw similar trends in *w*MelM mosquitoes, where mosquitoes with a disseminated infection had higher levels of DENV-2 (Fig. S2a). As with *w*AlbB mosquitoes, there was no difference in *Wolbachia* density between *w*MelM mosquitoes that were either not infected, were infected but had a non-disseminated infection, or had a disseminated infection (Fig. S2b). Thus, increased dissemination rates were associated with increased DENV-2 levels, absent large differences in *Wolbachia* density, perhaps due to earlier escape from the midgut, and potentially resulting in more opportunities for replication throughout the mosquito body.

### Successive feeding results in earlier dissemination

In our previous studies, we found that multiple blood meals not only increased dissemination rates but also shortened the EIP[10,16]. Further, we demonstrated that forced salivation assays were less accurate at predicting transmission than using the presence of virus in mosquito legs (dissemination) as a proxy for transmission ability[24]. To determine if a similar temporal shift in dissemination and subsequent EIP occurs in *Ae. aegypti* mosquitoes transinfected with *w*AlbB *Wolbachia*, we conducted time course assays examining infection and dissemination in single- and double-fed WT and *w*AlbB mosquitoes across days 5–10 post-infectious blood meal (Fig. 3a). We did not include *w*MelM mosquitoes in these assays due to the very low rate of infection and dissemination observed in initial experiments despite substantial numbers (Fig. 1b, c). Our experiments with WT mosquitoes resulted in similar infection rates over time between groups (Fig. 3b) and significantly earlier dissemination (i.e., shorter EIP) in the double-fed group as compared to the single-fed group, as we previously observed with other virus-vector pairings (Fig. 3c)[10,24]. Importantly, we observed similar patterns for infection and dissemination in mosquitoes with *w*AlbB, with similar infection rates between single- and double-fed groups at all timepoints (Fig. 3d), and a shorter EIP due to increased dissemination during the earlier timepoints (Fig. 3e). When we examined *Wolbachia* density over time in single- and double-fed mosquitoes, we did not observe any clear differences in relative *w*AlbB densities in bodies (Fig. S3a) or midguts (Fig. S3b) at any of the timepoints following blood feeding. In summary, these results are consistent with our previous studies, indicating that successive feeding leads to earlier dissemination from the midgut and suggests that the presence of *w*AlbB *Wolbachia* does not disrupt this phenotype[10,16].

### Successive feeding shortens the time until 50% dissemination (EIP$_{50}$)

To measure the impact of successive feeding on timing of dissemination, we modeled time to dissemination using empirical data from our time course experiments (Fig. 4). To quantitatively assess the differences between groups, we then estimated the time at which 50% of WT and *w*AlbB mosquitoes had disseminated infection (EIP$_{50}$; Fig. 4). As described in the methods section, we used survival models assuming gamma-distributed dissemination times in which the shape (α), and rate (β) of the gamma distribution of DENV-2 dissemination might differ as a function of *w*AlbB infection status (*w*) and blood-feeding status (*f*). As an initial exploration, we fitted four different models: 1) one with four sets of α$_{w,f}$, and β$_{w,f}$ parameters for each combination of *w* and *f*; 2) one with two sets of α$_w$ and β$_w$ parameters for each *w*; 3) one with two sets of α$_f$ and β$_f$ parameters for each *f*; and 4) one with a single set of α and β parameters. When these models were compared in a pairwise fashion using Bayes factors, the first model that used different dissemination time distributions for all four types of mosquitoes (*w*AlbB/WT × SF/DF) fit the data best, with a Bayes factor of BF$_{1>2}$ = 2.3 × 10$^4$ when compared to the second-best model (Table S1). Additional comparisons between models and graphs of prior and posterior shape and rate parameters can be found in supplemental data (Table S1 and Fig. S4). From the model of best fit, we calculated the number of days it would take for 50% of mosquitoes to develop a disseminated infection (EIP$_{50}$). As 50% dissemination was exceeded at every timepoint examined in WT mosquitoes, our modeling was not able to determine a day when 50% of WT mosquitoes would have a disseminated infection (Fig. 4a, b). For single-fed *w*AlbB mosquitoes, 50% dissemination was reached at 8.38 days post-infection (95% credible interval [CrI]: 7.72–9.01; Fig. 4c). Double-fed *w*AlbB mosquitoes reached 50% dissemination earlier at 6.86 days post-infection (95% credible interval [CrI]: 6.03–7.62; Fig. 4d).

### Lower odds of *w*AlbB surviving the EIP relative to wildtype mosquitoes

Using our model predictions of dissemination time as an estimate of EIP, we went on to predict the probability of mosquitoes surviving past the EIP given varying average lengths of mosquito lifespan (Fig. 5a). In all instances, double-fed WT or *w*AlbB mosquitoes were more likely to survive beyond the EIP than single-fed counterparts (Fig. 5a). WT mosquitoes were more likely to survive beyond the EIP than *w*AlbB mosquitoes, regardless of number of blood meals, indicating that *w*AlbB reduces the likelihood of DENV-2 dissemination (Fig. 5a). In general, the probability of mosquitoes surviving beyond the EIP increased as average mosquito lifespan increased; however, WT double-fed mosquitoes were highly likely to survive beyond the EIP regardless of average mosquito lifespan (Fig. 5a). We went on to quantify the epidemiological significance of these factors by calculating the associated odds ratio of surviving past the EIP, associated with *w*AlbB infection status and how that effect was modulated by blood-feeding status (Fig. 5b). This comparison is particularly relevant as it enables us to determine the odds ratios for surviving the EIP for *w*AlbB relative to WT mosquitoes, under both the traditional single-fed and more natural and epidemiologically relevant double-fed scenarios. For both single- and double-fed mosquitoes, the odds ratio of surviving beyond the EIP increased as average mosquito lifespan increased, indicating that mosquitoes with *w*AlbB were less likely to survive long enough to transmit DENV-2 when lifespan was shorter (Fig. 5b). This is due to the increase in time to dissemination observed in *w*AlbB mosquitoes relative to WT mosquitoes and reflects the interaction between virus dissemination and mosquito mortality. The odds ratios for double-fed mosquitoes were much smaller than those of single-fed mosquitoes, indicating that although successive feeding did reduce EIP for *w*AlbB mosquitoes, successive

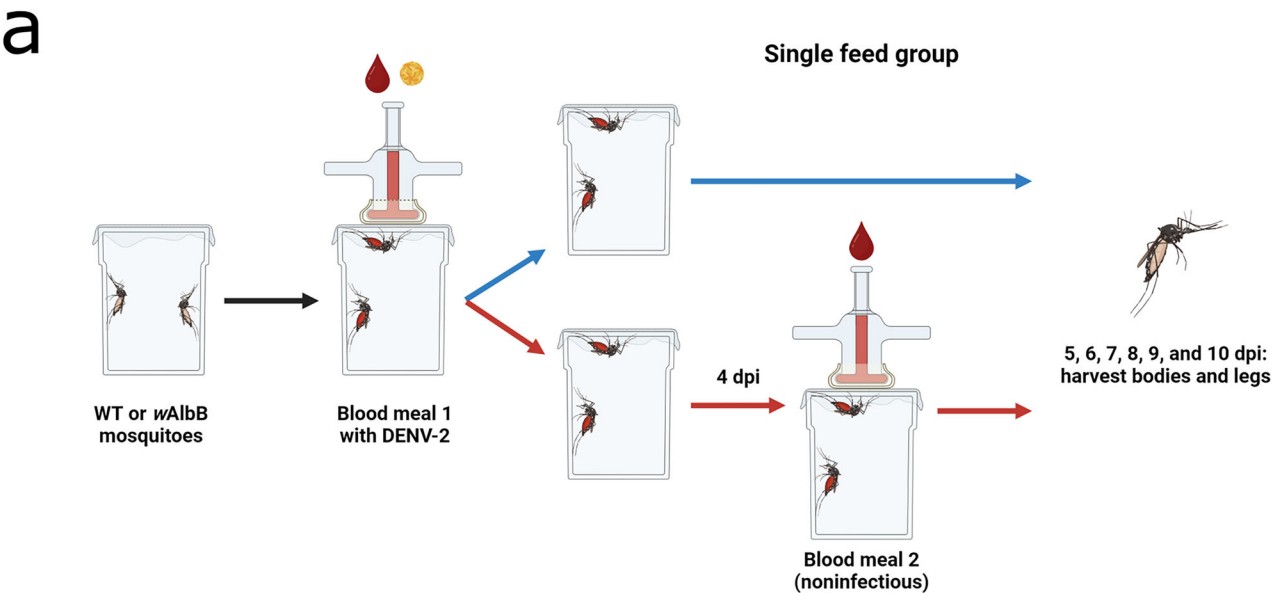

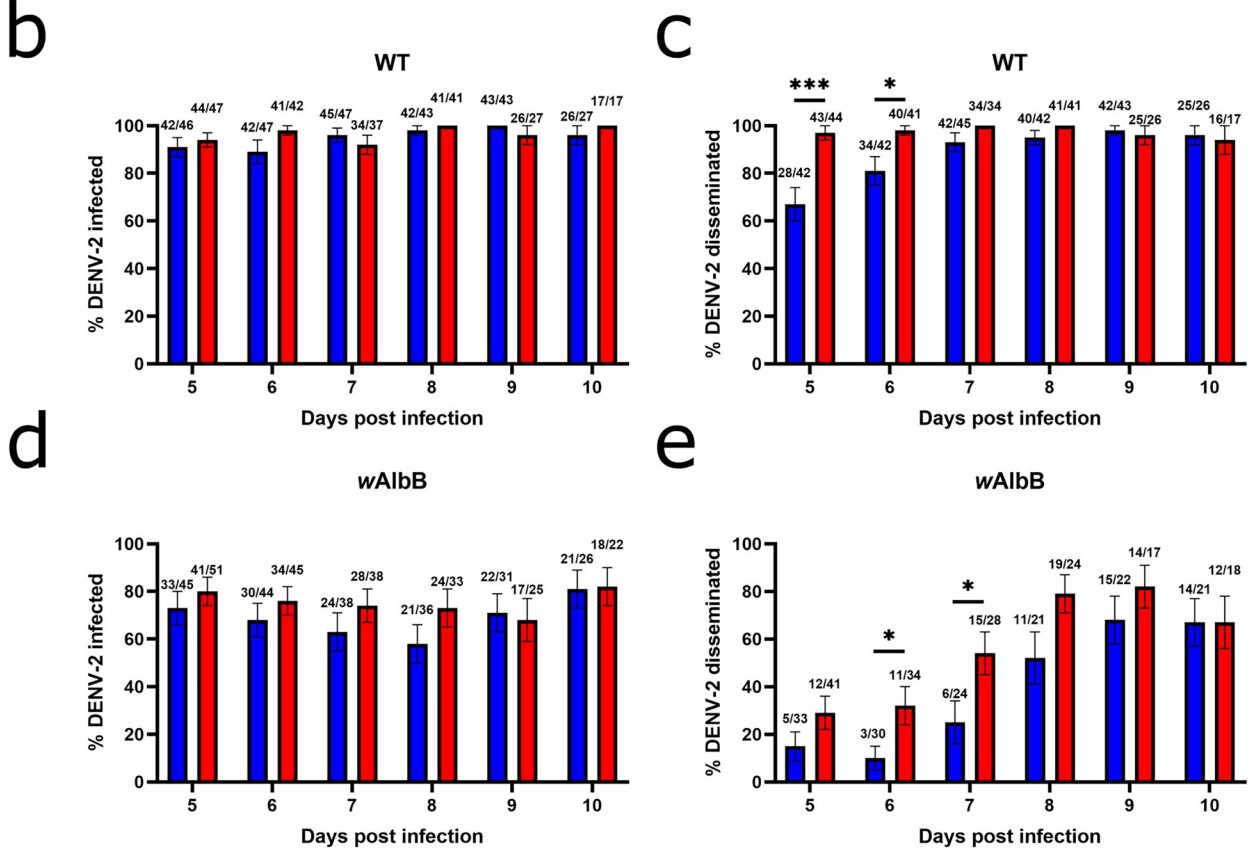

feeding has a larger impact on EIP in WT mosquitoes (Fig. 5b). This suggests that *w*AlbB remains effective in inhibiting DENV-2 when considering successive feeding.

## Discussion
This work provides important new insights into how mosquito blood feeding behavior impacts the efficiency of *Wolbachia*-based

DENV-2 inhibition. In this study, we expanded on our prior work and demonstrated increased early DENV-2 dissemination and shorter EIP in both the presence and absence of *w*AlbB *Wolbachia* when *Ae. aegypti* mosquitoes were given a second non-infectious blood meal. By modeling the odds ratio of surviving beyond the EIP, we show that the impact of successive feeding is larger on WT mosquitoes (wildtype mosquitoes lacking *Wolbachia*) as

**Fig. 3 | Successive feeding accelerates DENV-2 dissemination and shortens the extrinsic incubation period in both WT and *w*AlbB *Wolbachia*-transinfected *Ae. aegypti*. a** Experimental design for infection and dissemination time course studies of single- and double-fed wildtype mosquitoes lacking *Wolbachia*. *Created in BioRender. Brackney, D. (2025)* https://BioRender.com/53u2wpv (WT) and *w*AlbB mosquitoes. **b** Percentage of infected single- and double-fed WT mosquitoes 5–10 dpi. Numbers indicate infected mosquitoes over total fed mosquitoes. **c** Percentage of single- and double-fed WT mosquitoes with disseminated infection 5–10 dpi. Numbers indicate mosquitoes with disseminated infection (measured using legs) over infected mosquitoes. WT single-fed vs double-fed day 5 p = 0.0001. WT single-fed vs double-fed day 6 p = 0.0294. **d** Percentage of infected single- and

double-fed *w*AlbB mosquitoes 5–10 dpi. Numbers indicate infected mosquitoes over total fed mosquitoes. **e** Percentage of single- and double-fed *w*AlbB mosquitoes with disseminated infection 5–10 dpi. *w*AlbB single-fed vs double-fed day 6 p = 0.0378. *w*AlbB single-fed vs double-fed day 7 p = 0.0494. Numbers indicate mosquitoes with disseminated infection (measured using legs) over infected mosquitoes. Comparisons were made using two-sided Fisher's exact tests. *p ≤ 0.05, **p ≤ 0.01, ***p ≤ 0.001, ****p < 0.0001. Blue = single-fed, red = double-fed. Lines indicate mean ± standard error of the mean of the total sample proportions. Data for time course experiments were collected over 2 replicates. Source data for (**b**–**e**) are provided as a Source Data file.

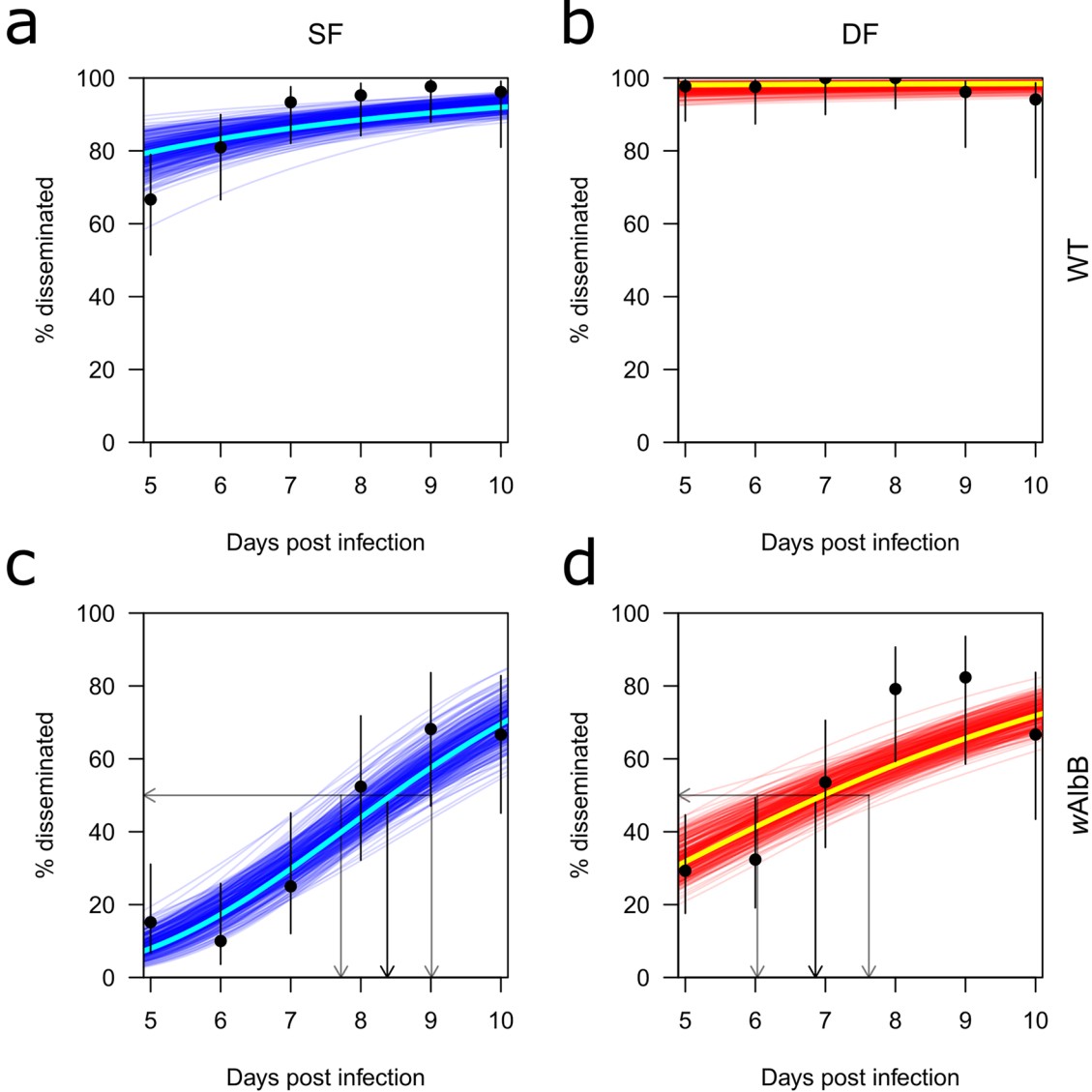

**Fig. 4 | Modeling of timing of dissemination in single and double-fed WT and *w*AlbB *Wolbachia* mosquitoes.** Modeling of timing of dissemination using empirical data shown in Fig. 3 for **a** single-fed WT mosquitoes (wildtype mosquitoes without *Wolbachia*), **b** double-fed WT mosquitoes, **c** single-fed *w*AlbB mosquitoes, and **d** double-fed *w*AlbB mosquitoes. Experimental data are shown as black dots, with black lines indicating the raw uncertainty in the proportions disseminated. Dark blue or red lines show different draws from the posterior distribution of parameters and indicate

the model's uncertainty. Bright blue and red lines represent the model's maximum *a posteriori* prediction. Black lines with arrows mark the timing of 50% dissemination (EIP$_{50}$) with 95% credible intervals marked by flanking gray lines with arrows. For (**c**) 50% dissemination = 8.38 days post-infection (95% credible interval [CrI]: 7.72–9.01. For (**d**) 50% dissemination = 6.86 days post-infection (95% credible interval [CrI]: 6.03–7.62. Blue = single-fed, red = double-fed. SF = single-fed, DF = double-fed. Data used to model the dissemination time course was collected over 2 replicates.

compared to *w*AlbB mosquitoes. This suggests that *Wolbachia* remains an effective strategy to inhibit DENV-2 transmission even under successive feeding conditions and that traditional single-feeding experiments that fail to account for *Ae. aegypti* successive

feeding behavior may underestimate the effectiveness of *w*AlbB *Wolbachia*.

We also found that WT mosquitoes were much more susceptible to DENV-2 infection than mosquitoes with either *w*MelM or *w*AlbB,

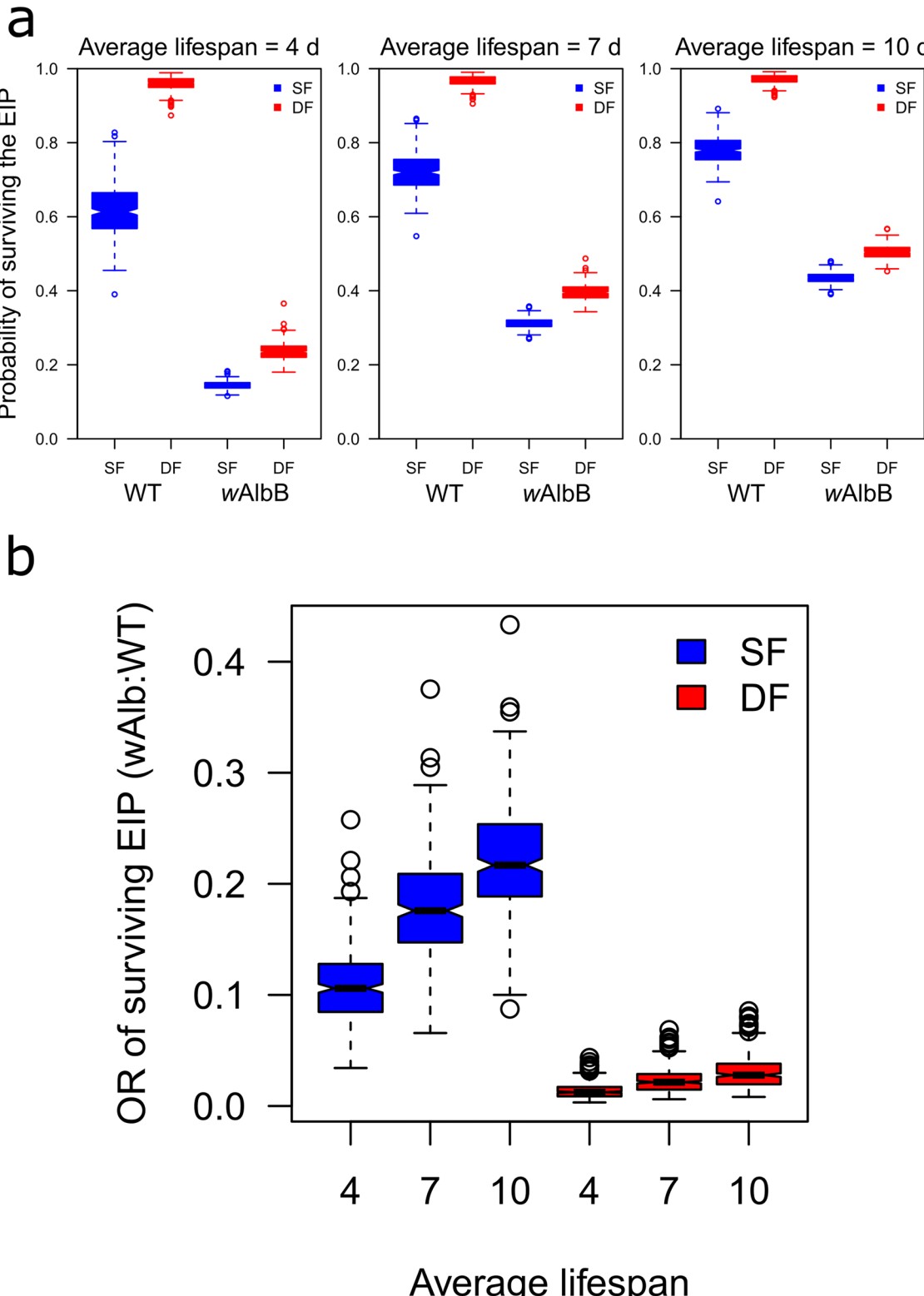

**Fig. 5 | Impact of feeding behavior on the probability of a mosquito surviving beyond the extrinsic incubation period (EIP). a** Probability of single- and double-fed wildtype mosquitoes without *Wolbachia* (WT) and *w*AlbB mosquitoes surviving past the EIP assuming an average mosquito lifespan of 4, 7, or 10 days. **b** Odds ratio of surviving past the EIP (*w*AlbB:WT) given a mosquito lifespan of 4, 7, or 10 days. Center lines indicate median posterior predictions. Boxes and whiskers indicate 50% and 95% posterior prediction intervals, respectively. Circles indicate posterior samples falling beyond the 95% posterior prediction intervals. Blue = single-fed, red = double-fed. SF = single-fed, DF = double-fed.

indicating strong virus inhibition with *Wolbachia* as has been shown in previous studies (Fig. 1b)[3,18,25,26]. Further, fewer mosquitoes with *w*MelM became infected than those with *w*AlbB, indicating that *w*MelM provides stronger DENV-2 inhibition or that more of the inhibitory effects of *w*MelM occur prior to DENV-2 infection of the midgut (Fig. 1b). For all groups, infection rates were largely unchanged by feeding behavior (Figs. 1b and 3b, d), however, mosquitoes with *w*MelM that were given a second blood meal had a slight increase in infection rate that may be an artifact of the low number of mosquitoes infected (Fig. 1b). Time course experiments revealed that both WT and *w*AlbB mosquitoes had higher rates of early dissemination when given a second blood meal (Figs. 1c and 3c, e). This is in line with earlier results from WT mosquitoes and indicates that, despite viral inhibition by *w*AlbB, earlier DENV-2 dissemination in mosquitoes with *w*AlbB *Wolbachia* is facilitated by additional non-infectious blood meals[10,13].

Interestingly, we found that when WT and *w*AlbB mosquitoes were given a second blood meal, there was an increase in DENV-2 levels in the mosquito body (Fig. 2a). In agreement with a previous study, we found little variation in *w*AlbB or *w*MelM *Wolbachia* levels by feeding or infection or dissemination status (Figs. 2b, d, S1a–d, S2b and S3a, b)[19]. Thus, we propose that the differences in virus titer we observed are not correlated with *Wolbachia* density, as has previously been suggested[27]. When DENV-2 levels in *w*AlbB and *w*MelM mosquitoes were compared by dissemination status (Figs. 2c and S2a), mosquitoes with disseminated infections had higher levels of DENV-2, which may indicate that midgut escape allows for invasion of new tissues and increased virus replication or that mosquitoes with higher DENV-2 levels are more likely to become disseminated, as seen in previous studies[16].

When dissemination time course data from WT and *w*AlbB mosquitoes was used to predict the time needed for 50% of mosquitoes to reach a disseminated infection ($EIP_{50}$), we estimated that the incubation periods was ~2 days shorter in double-fed as compared to single-fed *w*AlbB mosquitoes (Figs. 3e and 4c, d). Experimentally, differences were seen between dissemination rates in WT single- and double-fed mosquitoes, though predictions of 50% dissemination timing were not possible in WT groups as dissemination was always higher than 50% across the timeframe examined (Figs. 3c and 4a, b, and Table S2). This is likely due to the starting DENV-2 titer used in our experiments and represents a limitation of our study, but also reflects the strong inhibition of infection and dissemination seen in *Wolbachia*-transinfected mosquitoes relative to highly susceptible WT populations (Table S2). Predictions of the probability of mosquitoes surviving past the EIP further reinforced this dynamic, as WT mosquitoes from both single- and double-fed groups were more likely to survive beyond the EIP than their *w*AlbB counterparts given any plausible mosquito lifespan (Fig. 5a). Although successive feeding always increased the probability of surviving beyond the EIP, when comparing the odds ratio of *w*AlbB:WT mosquitoes surviving past the EIP, it became evident that successive feeding increases the probability of WT mosquitoes surviving past the EIP more than it does in mosquitoes with *w*AlbB (Fig. 5b). Given this, *Wolbachia* may have an even stronger impact than previously thought when the sequential feeding behavior of *Ae. aegypti* in the wild is taken into consideration. This lends further support to releases of mosquitoes transinfected with *Wolbachia* in the field, and may help explain the reductions in DENV observed in several countries with *Wolbachia*-transinfected mosquito releases[6,28]. Despite this promising data, it is worth noting that our data is primarily from mosquitoes with *w*AlbB, and further study of mosquitoes with different strains of *w*Mel is warranted. Additionally, this model does not consider other indirect impacts of *Wolbachia* on mosquito life history traits such as lifespan, fecundity, and feeding frequency[29–32].

While these findings represent valuable new information regarding virus dynamics in mosquitoes with *w*AlbB *Wolbachia*, our study has some limitations. First, although this study attempts to provide a more

accurate model of mosquito behavior and feeding in the wild, laboratory experiments are inherently artificial and, in this case, relied on water-jacketed membrane feeders and defibrinated sheep blood rather than a live host with a functioning immune system. We also provided mosquitoes with complete blood meals, whereas mosquitoes in the wild frequently take partial blood meals, and the viral titers we used to infect mosquitoes may be higher than those often encountered in the wild[15,33]. Additionally, many of our findings are based on data from *w*AlbB mosquitoes as *w*MelM was highly efficient at disrupting DENV-2 infection and subsequent dissemination, and we were not able to detect a difference in dissemination rates between single- and double-fed *w*MelM groups due to low numbers (Fig. 1c). Despite this, the observed trends in *w*AlbB mosquitoes are important to take into account when modeling or considering *Wolbachia*-based interventions and should be examined further using other *Wolbachia* strains and host genetic backgrounds[34]. One finding that should be addressed is the low percentage of *w*AlbB mosquitoes with detectable DENV-2 in their saliva when examined 10 dpi, despite adequate levels of infection and dissemination and elevated DENV-2 levels with double-feeding as observed in experiments conducted at earlier timepoints (Figs. 1b, c, 2a, 3d, e and S5a–e). These assays used forced salivation techniques and, although detection of virus in mosquito saliva can be used to measure transmission ability, previous work has suggested that such artificial salivation assays often underestimate transmission ability and that dissemination, as measured by taking mosquito legs, more closely reflects the ability of a mosquito to pass on infection[24]. As such, we used dissemination as an estimate of EIP and trust these findings over the limited results from the forced salivation assays.

Despite these limitations, our work provides important new insights into the impact of mosquito feeding behavior on DENV-2 inhibition by *Wolbachia* that will be valuable for future modeling and control efforts. Most prior studies of *Wolbachia*-mediated virus inhibition have not considered the tendency of *Ae. aegypti* to feed frequently and have instead relied on experiments using a single infectious blood meal. Our results indicate that successive blood feeding can impact DENV-2 dissemination timing and the subsequent probability of both WT and *w*AlbB *Wolbachia*-transinfected mosquitoes surviving beyond the EIP. While our work found increased dissemination with successive feeding in WT mosquitoes and those with *w*AlbB, we also found robust DENV-2 inhibition by *Wolbachia* in both *w*AlbB and *w*MelM groups, with *w*MelM exhibiting even stronger inhibition than *w*AlbB. Our modeling suggests that functional *w*AlbB inhibition of DENV-2 may be even stronger than previously thought due to the larger impact of successive feeding on EIP survivability in WT mosquitoes when compared to *w*AlbB mosquitoes. These results stress the importance of considering mosquito behavior when designing laboratory experiments or modeling control efforts and provide a clearer understanding of DENV-2 infection dynamics in mosquitoes with *Wolbachia* under single- and successively fed conditions.

## Methods
### Mosquito rearing
*Ae. aegypti* mosquito lines (WT, *w*MelM, and *w*AlbB) were generated from natively uninfected *Ae. aegypti* collected near Cairns, Queensland, Australia, that were kept uninfected (WT; wildtype mosquitoes without *Wolbachia*) or transinfected with either *Wolbachia w*AlbB or *w*MelM strain via microinjection[35,36]. All three mosquito colonies were maintained in separate environmental chambers at 26 °C, 60–70% relative humidity, and a 12:12 light-dark cycle. Larvae were hatched from egg papers in 500 mL of water and two drops of Liquifry No. 1 fish food. After hatching, ~250 first-instar larvae were transferred to trays with 1 L of water and fed with Tetramin baby fish food. Pupae were collected and transferred to Bugdorm-1 cages for adults to emerge. Adult mosquitoes were maintained on 10% sucrose and blood-fed with

defibrinated sheep blood (HemoStat Laboratories). Adults that were ~1 week old were sorted into cups of 60 female mosquitoes/cup and maintained on 10% sucrose-soaked pads prior to and following infection with DENV-2.

## Dengue virus

DENV-2 (125270/VENE93; GenBank: PQ852084) was grown in *Ae. albopictus* C6/36 cells in T75 flasks and split at a 1:15 dilution in 10% FBS MEM media. For DENV-2 infections, when cells were 60-80% confluent, growth media was removed and a thawed 250 μL aliquot of DENV-2 was added to flasks along with 3 mL of 10% FBS MEM media. Flasks were placed on a rocking platform for 1 h before 12 mL of 10% FBS MEM media was added for a total volume of 15 mL. Infected cells were grown in a 28 °C incubator with 5% $CO_2$ for 5 days. DENV-2 was harvested by removing the supernatant from cells 5 days post-infection (dpi). Dilutions of virus-containing cellular supernatant in defibrinated sheep blood were fed to mosquitoes during the primary infectious blood meal (Table S2).

## Mosquito infections, blood feeding, and stock virus quantification

One day prior to the infectious blood meal, sucrose-soaked pads were replaced with water-soaked pads to stimulate blood feeding. Mosquitoes were fed with 1:5 or 1:12 dilutions of DENV-2 and defibrinated sheep's blood (Table S2). Virus titers fed to mosquitoes were quantified by both RT-qPCR and focus-forming assay as described previously (Table S2)[12]. Mosquitoes were knocked down on ice, and blood-fed mosquitoes were sorted into containers and provided with an oviposition cup. Mosquitoes in the double-feed group were given a second, non-infectious blood meal of defibrinated sheep's blood 4 days after the initial infectious blood meal (Figs. 1a, 3a and S5a). Blood-fed mosquitoes were sorted into new containers and given an oviposition cup. Mosquitoes in both the "single-feed (SF)" and "double-feed (DF)" groups were sacrificed at different days, ranging from 5 to 10 days post-infectious blood meal.

## Mosquito tissue collections and extractions

For initial experiments assessing infection and dissemination differences between SF and DF groups at 7 dpi (Infection and dissemination rep 1–5), mosquito bodies were separated from legs and wings (Table S2). To assess infection, each mosquito body was placed in a separate 2 mL tube with 200 μL of mosquito diluent (1X phosphate buffered saline with 20% heat-inactivated fetal bovine serum, 50 μg/ml penicillin/streptomycin, 50 μg/ml gentamycin, and 2.5 μg/ml amphotericin B) and a copper bead. To assess dissemination, legs and wings were pooled from individual mosquitoes in a 2 mL tube containing 200 μL of mosquito diluent and a copper bead.

For time course experiments comparing the shift in timing of dissemination (Time course rep 1–2), mosquito bodies and legs were harvested at 5–10 dpi (Table S2). As before, tissues were collected into tubes containing 200 μL of mosquito diluent and a copper bead.

To examine salivary transmission in *w*AlbB transinfected mosquitoes (Salivation experiments wAlbB), salivation assays were conducted at 10 dpi (Table S2). Wings and legs were pooled in tubes containing mosquito diluent as before, and saliva was collected by placing the proboscis of each incapacitated mosquito into a 20 μL pipette tip with 5 μL of a 50:50 mix of 50% sucrose and FBS. Mosquitoes were allowed to salivate for 1 h before bodies were harvested as before, and the saliva-containing solution was expelled into a 2 mL tube containing 100 μL mosquito diluent and a copper bead.

All samples were stored at −80 °C until homogenization using a Retsch Mixer Mill 400 for 4 min at 30 Hz, followed by centrifugation for 5 min at 7000 rcf. Nucleic acid was extracted from homogenate (75 μL) using the ThermoFisher MagMAX viral/pathogen nucleic acid isolation kit and eluted into 75 μL using the KingFisher Flex system.

## DENV RT-qPCR and FFA

Mosquito samples were screened for DENV-2 RNA using the NEB Luna Universal Probe One-Step RT-qPCR Kit on the Bio-Rad CFX-96 touch real-time PCR detection system using previously developed primers (Table S3)[37]. PCR conditions were as follows: 55 °C for 10 min, 95 °C for 1 min, and 40 cycles of 95 °C for 10 s followed by 55 °C for 30 s and a plate read. All plates were run with at least one negative extraction control, one negative template control, and a serial dilution of synthetic RNA transcript. All samples that were quantified were run in duplicate on the same plate. Positivity was determined by Ct value; samples with a Ct value below 37 were considered positive.

A subset of mosquito bodies from WT and *w*AlbB mosquitoes (7 dpi) was used to compare DENV-2 genome equivalents per mL as determined via RT-qPCR to viral titers via focus-forming assay (FFA) (Fig. S6). For FFAs, *Ae. albopictus* C6/36 cells were seeded into 96-well plates at a density of $3 \times 10^5$ cells/well, incubated overnight at 28 °C with 5% $CO_2$, and infected the following day with 30 μL per well of virus-containing serially diluted mosquito sample for 1 h at 28 °C with 5% $CO_2$. Virus-containing supernatant was removed, cells were covered with 100 μL of 1% methylcellulose in 10% FBS MEM media, and cells were incubated for 3 days at 28 °C with 5% $CO_2$. Cells were then fixed for 15 min at room temperature with 100 μL of 4% formaldehyde in PBS, washed 3 times with 100 μL PBS, permeabilized with 0.2% Triton-X in PBS for 10 min at room temperature, washed again 3 times, and 30 μL of mouse anti-flavivirus group antigen antibody from NovusBio D1-4G2-4-15 (4G2) diluted 1:500 in PBS was added to each well. Plates were incubated overnight at 4 °C. Plates were then washed 3 times with PBS and then incubated overnight at 4 °C with 30 μL of Invitrogen goat anti-mouse IgG (H+L) cross-adsorbed secondary antibody, Alexa Fluor 488 diluted 1:200 in PBS. The following day, plates were washed to remove excess secondary antibody, and foci were counted using a Zeiss Axio Vert.A1 inverted microscope with a 2.5× objective and a FITC filter. As expected, DENV-2 genome equivalents/mL via RT-qPCR for body samples were higher than focus-forming units/mL via FFA, yet DENV-2 concentrations via each method were internally consistent and concentrations are correlated between both methods (Fig. S6). This provided justification for using RT-qPCR rather than FFA to measure DENV-2 concentration in experiments.

## *Wolbachia* and *Aedes* qPCR

Samples were screened for *Wolbachia* genome equivalents using primers targeting a DNA-directed RNA polymerase subunit beta/betagene with locus tag WD_RS06155 and compared to *Ae. aegypti* S6 DNA copies using the NEB Luna Universal qPCR kit on the CFX Connect Real-Time PCR Detection System, and protocols and primers were modified from Lau et al. and Lee et al. (Table S3)[38,39]. PCR conditions were as follows: 95 °C for 3 min, 40 cycles of 95 °C for 10 s and 60 °C for 30 s with a plate read at the end of each cycle. A melt curve was run from 65 °C to 95 °C at a rate of 0.5 °C every 5 s with a plate read every 5 s. All plates were run with at least one negative extraction control, one negative template control, and one each of confirmed positive RNA extracts from WT and *w*MelM, and *w*AlbB-infected colonies. All samples that were quantified were run in duplicate on the same plate. Positivity was determined by the peak of the melting curve. For the wMwA *Wolbachia* detection assay, WT mosquitoes produced no melt peak, *w*MelM presence resulted in a melt peak at 80.5 °C, and *w*AlbB produced a melt peak at 78–78.5 °C. *Ae. aegypti* DNA detection served as an *Ae. aegypti* genome copies control, and all mosquito groups produced similar melt peaks at 81.5–82.5 °C. Relative *Wolbachia* densities were determined by taking the average crossing point (Cp) of the *Wolbachia*-specific marker and the average Cp value of the *Ae. aegypti*-specific marker across 2 duplicate wells. The average *Wolbachia*-specific marker Cp was then subtracted from the average *Ae. aegypti*-specific marker Cp and transformed by $2^n$ as described before[23].

## Experimental data analysis

Comparisons of proportions of DENV-2 infection, dissemination, and saliva positivity were made using contingency analyses with two-sided Fisher's exact tests. DENV-2 and *Wolbachia* concentration differences between single- and double-fed groups were compared on untransformed data using two-tailed Mann-Whitney U tests. Two-tailed nonparametric Spearman correlation tests were used to test for correlation between DENV-2 and *Wolbachia* concentrations in individual mosquitoes. For comparisons between uninfected mosquitoes, mosquitoes with disseminated infections, and mosquitoes with non-disseminated infections, a Kruskal-Wallis test with Dunn's post hoc test for multiple comparisons was used on untransformed data. Specific statistical tests are noted in the legend for each graph. Graphs and statistical comparisons were made using GraphPad Prism 10.5.0. Experimental diagrams were made using BioRender.

## Analysis and modeling of results

To analyze the effect of feeding status and *Wolbachia* infection on time to dissemination, we performed a survival analysis assuming a gamma-distributed time to dissemination. This allowed the hazard rate to increase over time, consistent with the expectation that dissemination would be unlikely until after some minimum amount of time. More specifically, we modeled the dpi for DENV-2 infection to disseminate to the salivary glands, $D$, as a gamma-distributed random variable, which is defined by a shape parameter, $\alpha$, and a rate parameter, $\beta$. Because $D$ is a continuous random variable and dissemination status was recorded at a daily resolution, the probability that $D = d$ is $F(d;\alpha,\beta)$ for $d = 5$ and $F(d;\alpha,\beta) - F(d-1;\alpha,\beta)$ for $d \in \{6,7,8,9,10\}$, where $F(d;\alpha,\beta)$ is the gamma cumulative distribution function. Out of $N_d$ mosquitoes tested for dissemination on day $d$, $X_d$ were positive. Together, this means that $\Pr(X_d = x) = \mathrm{Binomial}(x;N_d,\Pr(D = d))$, where Binomial refers to the probability mass function of a binomial random variable.

Batches of mosquitoes tested for DENV-2 dissemination were distinguished by their *w*AlbB infection status, $w$, and their blood-feeding status, $f$. We considered the possibility that $\alpha$ and $\beta$ might differ as a function of $w$ and $f$. To assess this, we fitted four different models: 1) one with four sets of $\alpha_{w,f}$, and $\beta_{w,f}$ parameters for each combination of $w$ and $f$; 2) one with two sets of $\alpha_w$ and $\beta_w$ parameters for each $w$; 3) one with two sets of $\alpha_f$ and $\beta_f$ parameters for each $f$; and 4) one with a single set of $\alpha$ and $\beta$ parameters. We compared these models in a pairwise fashion using Bayes factors, which were obtained by taking the ratio of the marginal likelihoods of two models using the marginalLikelihood function in the BayesianTools package in R version 4.3.2[40,41]. A Bayes factor of 10 or greater was taken as evidence of strong support of the model with the higher marginal likelihood[42].

The likelihood of each model was $L(\{\alpha_{w,f}\},\{\beta_{w,f}\}|\{X_{d,w,f}\}) = \Pi_{d,w,f} \Pr(X_d = x)$. To define priors for the parameters, we referred to two previous studies[10,17]. For $\alpha$ and $\beta$, we used previously determined joint posterior estimates of $\alpha_{\mathrm{WT,SF}}$, $\alpha_{\mathrm{WT,DF}}$, $\beta_{\mathrm{WT,SF}}$, and $\beta_{\mathrm{WT,DF}}$ to define our joint prior distribution for those parameters[10]. Consistent with previous estimates of the effect of *w*MelM on dissemination, we multiplied prior samples of $\alpha_{\mathrm{WT,SF}}$, and $\alpha_{\mathrm{WT,DF}}$ by samples from a normal distribution with a mean of 1.276 and a standard deviation of 0.1469 to obtain prior samples of $\alpha_{w\mathrm{AlbB,SF}}$, and $\alpha_{w\mathrm{AlbB,DF}}$[17]. Because the estimates from Ye et al.[17] only describe the effect of *w*MelM on mean time to dissemination, and the ratio of the means of two gamma distributions can be described by the ratio of their shape parameters only, we assumed the same prior distributions for $\beta_{w\mathrm{AlbB,SF}}$, and $\beta_{w\mathrm{AlbB,DF}}$ as we did for $\beta_{\mathrm{WT,SF}}$, and $\beta_{\mathrm{WT,DF}}$[17]. Due to a lack of prior knowledge about the correlation among shape and rate parameters for WT and *w*AlbB mosquitoes, we scrambled the ordering between these two sets of parameter samples to make their priors independent.

We obtained posterior estimates of the parameters of each model using the Metropolis sampler in the BayesianTools package in R

version 4.3.2[40,41]. We used three chains totaling 105 iterations. We assessed convergence of these chains through visual inspection of trace plots and confirmation that parameter-wise and multivariate Gelman-Rubin diagnostics were all indistinguishable from 1 using the gelmanDiagnostics function in the BayesianTools package in R version 4.3.2[40,41].

To assess the epidemiological significance of our estimates, we used model predictions of dissemination time to calculate model predictions of the probability of a mosquito surviving long enough for virus to disseminate to the legs or Pr(survive to disseminate). The closely related probability of surviving past the EIP is a key quantity in the canonical Ross-Macdonald theory of mosquito-borne pathogen transmission, and our earlier work has established that dissemination is an accurate predictor of transmission ability and thus EIP[24,43]. We focused on this metric because EIP is the only parameter in the Ross-Macdonald expression for the basic reproduction number, $R_0$, affected by blood-feeding status, and it is also expected to be strongly affected by *w*AlbB infection status[43]. Thus, the relative effect of blood-feeding status and *w*AlbB infection status on Pr(survive beyond the EIP) is identical to their relative effect on $R_0$. Because dissemination time is a random variable in our analysis, we calculate an expected value of Pr(survive to disseminate) as

$$E[\Pr(\text{survive to disseminate})] = \int e^{-D/L}\Pr(D)dD, \qquad (1)$$

Where $D$ is dissemination time, $\Pr(D)$ is the gamma distribution for $D$ estimated above, and $L$ is average mosquito lifespan (we explored values of 4, 7, and 10 days). In addition to this quantity, we also calculated the epidemiological significance of blood-feeding status and *w*AlbB infection status by calculating the associated odds ratio, OR(survive to disseminate), associated with *w*AlbB infection status and how that effect was modulated by blood-feeding status. Specifically, we calculated the odds ratio as

$$\mathrm{OR(\text{survive to disseminate})} = \frac{E[\Pr(\text{survive to disseminate}]}{1 - E[\Pr(\text{survive to disseminate})]}. \qquad (2)$$

The advantage of an odds ratio for quantifying the effect of *w*AlbB on surviving the EIP is that it accounts for differences in the baseline probability of surviving the EIP for SF and DF mosquitoes.

## Reporting summary

Further information on research design is available in the Nature Portfolio Reporting Summary linked to this article.

# Data availability

All data are included in this manuscript and the supplementary files. Source data are provided with this paper.

# Code availability

Custom code is available on GitHub at https://github.com/TAlexPerkins/doubleFeedWolbachia.

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

## Acknowledgements

We would like to thank Prof. Ary Hoffmann for providing resources critical for mosquito colony production. This publication was made possible by CTSA Grant Number UL1 TR001863 from the National Center for Advancing Translational Science (NCATS), a component of the National Institutes of Health (NIH), awarded to C.B.F.V., the National Institute of Allergy and Infectious Diseases of the NIH under award number AI148477 (R.M.J. and D.E.B.), the NIH T32AI055403 (A.S.), the National Science Foundation Graduate Research Fellowship under Grant No. DGE-2139841 (A.S.), NIH National Institute of General Medical Sciences

R35 MIRA program grant number R35GM143029 (T.A.P.), Richter Fellowship from Trumbull College, Yale University (B.L.N.), and the Ambrose Monell Foundation (C.B.F.V.). P.A.R. was supported by an Australian Research Council Discovery Early Career Researcher Award DE230100067 funded by the Australian Government. The contents of this work are solely the responsibility of the authors and do not necessarily represent the official views of NIH.

## Author contributions

R.M.J., N.D.G., T.A.P., D.E.B., and C.B.F.V. designed the study. P.A.R. and X.G. created transinfected *Wolbachia* (*w*AlbB and *w*MelM strain) mosquito colonies. R.M.J., M.I.B., B.L.N., A.S., I.M.O., and C.B.F.V. performed the experimental studies. R.M.J. and T.A.P. analyzed the data. R.M.J., T.A.P., D.E.B., and C.B.F.V. wrote the manuscript. All authors read, reviewed, and approved the manuscript.

## Competing interests

The authors declare no competing interests.
