## [Peer Review file · Nature Communications]

Implications of successive blood feeding on *Wolbachia*-mediated dengue virus inhibition in *Aedes aegypti* mosquitoes

Corresponding Author: Dr Chantal Vogels

Version 0:

Reviewer comments:

Reviewer #1

(Remarks to the Author)

This study follows on from previous work of this group demonstrating that multiple feeding leads to increased dissemination of virus. Here they include *Wolbachia* infected mosquitoes, specifically a comparison between wMel and wAlbB. With the broader field grappling with which strain to release, these data provide useful additional data. They show that the blocking ability of wMel is strong enough to overpower any effect of double feeding on dissemination. The same is not true for wAlbB. They also showed that double feeds reduced EIP for WT and wAlbB infected mosquitoes (wMel not included given low numbers of virus infected). They then modeled average # of days to dissemination for the different treatment groups (WT + wAlbB x 2 feeding scenarios). Using the day of 50% of population with dissemination as an estimate of EIP, they then assessed the likelihood of mosquitoes surviving to this critical age point. Interestingly, double fed mosquitoes (WT and wAlbB) were more likely to survive. The difference in survival was more pronounced for WT because of the reduced viral loads due to blocking in wAlbB. In general, the likelihood of surviving EIP was much less for wAlbB. This is a solid piece of work that shows that blocking (*Wolbachia*) is generally more powerful than repeat feeding.

Methods

These vector competence + DENV load and + *Wolbachia* load assays are standard practice. The stats are robust.

Results

The graphics are clear.

Discussion

The authors are self-aware about how the experimental design may have affected their findings.

Suggestions

I find the use of the phrase 'survive EIP' to be awkward. I think you mean survive to EIP or beyond EIP. As written, it sounds like EIP is something to be survived....like an infection.

(Remarks on code availability)

Reviewer #2

(Remarks to the Author)

This manuscript by Johnson et al investigates the role of successive bloodmeals on the efficacy of *Wolbachia* in blocking DENV2 transmission. The paper is well written and the methodology is solid. I applaud the authors for investigating the multiple processes that EIP and dissemination feed into. I have mostly minor comments.

Major comment: Given that we've seen success of *Wolbachia* programs like World Mosquito Program, how does this jive

with your data? This is worth some Discussion.

Minor comments.

Lines 75-77 – are there a control group of the Wolbachia fed group? WT = not transfected? This isn't clear.

Does wild type mean from the field, or does it just mean non-transfected? If it is the latter, I think "control" might be a better label to avoid confusion. (I see where it's defined in the methods, but as those are at the end, "control" might be clearer anyway.)

Lines 81-83: this conclusion is not correct as broadly stated – you did not find a decrease in dissemination, for example, for "Wolbachia", but for one strain. This needs to be clarified throughout to be more specific.

Line 118-119: your concluding hypothesis here should probably be moved to the Discussion so as not to be misinterpreted as dogma

Line 135-140: Did you also assay saliva in addition to legs? Would be great to replicate that particular finding.

Lines 151-153: Given the lack of statistical significance, consider softening this conclusion - the next section supports this conclusion better. Also, EIP50 is usually the label given to this (just a consideration for ease of reading). Also, given your low rates in the one group, consider looking at EIP_{min} – also a good comparison as quicker dissemination even at low levels can affect transmission (see Mayton et al 2020)

Line 207-208 – I think this sentence is incomplete – reflects the interaction of dissemination and mortality?

Line 210-212: move to Discussion

Line 410: statistical

(Remarks on code availability)

Reviewer #3

(Remarks to the Author)

Summary:

The manuscript investigates the relationship between successive bloodmeals and virus dissemination in Wolbachia-infected mosquitoes, advancing the state of knowledge from several cited references (e.g., Amuzu et al., 2015, Armstrong et al., 2020; Brackney et al., 2021; Johnson et al., 2023). Specifically, the paper tests the effect of a second non-infectious bloodfeed on the effectiveness of Wolbachia-based inhibition of DENV-2 in *Aedes aegypti* mosquitoes in terms of DENV-2 infection and dissemination rates.

A statistical model of the timing of DENV-2 dissemination demonstrates that a second bloodfeed shortens EIP length by 1 day in mosquitoes carrying wAlbB. A second statistical model on the probability of mosquito survival over the EIP to show that a second blood feed increases this probability for both wildtype and wAlbB-infected mosquitoes, but to a much lesser degree for the latter. The authors conclude that models estimating the efficiency of Wolbachia blocking based on traditional single-feeding experiments are underestimating Wolbachia effectiveness.

Strengths:

The use of statistical modelling to translate classical vector competence measures into epidemiologically meaningful parameters such as EIP and probability of mosquito survival provided valuable insight.

Weaknesses:

- The results reported in the manuscript from Figures 1 to Figure 5A seem to present lower Wolbachia effectiveness and therefore worse epidemiological outcomes when double-fed wildtype or wAlbB mosquitoes are compared to their single-fed counterparts. It is surprising that the authors then claim that transmission models that take into account successive bloodfeeding would indicate higher Wolbachia effectiveness, which seems to be largely based on Figure 5B. A clearer description of how the odds ratios are calculated, and more explanation behind this rationalisation is needed.
- In this study, double-fed wAlbB mosquitoes are always compared to single-fed wAlbB but never to single-fed wildtype mosquitoes. The authors would probably find that double-fed wAlbB vs. single-fed wildtype mosquitoes show reduced Wolbachia effectiveness relative to single-fed wAlbB vs. single-fed wildtype mosquitoes. This is also an epidemiologically relevant comparison. The odds ratio calculations, for example, should take this into account too.
- There is not enough novelty nor mechanistic insight in the experimental findings to warrant publication in a broad-interest journal. It should be noted that these findings are not generalisable for all Wolbachia strains—the relevance of this manuscript is therefore limited to vector control efforts using the wAlbB. The strong point of the paper is in the transformation of vector biology empirical data into epidemiological significance—this work would be better suited for an epidemiology or vector biology journal. That said, the authors stopped short of performing transmission models themselves to demonstrate that the shortening of EIP truly translates to increased disease incidence.

Major comments:

- Line 165: Need some introduction as to why the 50% dissemination rate of WT and wAlbB mosquitoes is important to investigate.
- Line 242-245: That Wolbachia density was not affected by number of blood feeding has also been reported in Amuzu et al (2015). This would be an appropriate place to cite that work for this finding.
- Line 462-465: How is OR ratio calculated? Please rephrase for clarity.

- Figure 4c-d: Please, either through annotations on the graphs or in the figure legend, state the values of days post infections where the grey and black arrows fall.
- The number of replicates are only indicated in table S2. Please, add the information to the figure legend, including the number of mosquitoes analysed in each replicate.

Minor comments:

- Check for consistency in use of wMel and wMelM in text and figures
- Line 27: Correct to “Wolbachia transinfected mosquitoes”
- Line 199: The logical order of this sentence is inverted. WT mosquitoes are more likely to survive the WIP than wAlbB mosquitoes because wAlbB inhibits DENV-2 infection and dissemination. It would be more accurate to say “indicating that wAlbB reduces the likelihood of DENV-2 transmission”.
- Line 388: Indicate the gene used to quantify Wolbachia by qPCR.

(Remarks on code availability)

Version 1:

Reviewer comments:

Reviewer #3

(Remarks to the Author)

Overall, the authors made a sincere effort to correct all the reviewers' comments and corrections. They defended their argument well and have explained their rationale. I have nothing else to ask for.

(Remarks on code availability)

Reviewer 1

This study follows on from previous work of this group demonstrating that multiple feeding leads to increased dissemination of virus. Here they include Wolbachia infected mosquitoes, specifically a comparison between wMel and wAlbB. With the broader field grappling with which strain to release, these data provide useful additional data. They show that the blocking ability of wMel is strong enough to overpower any effect of double feeding on dissemination. The same is not true for wAlbB. They also showed that double feeds reduced EIP for WT and wAlbB infected mosquitoes (wMel not included given low numbers of virus infected). They then modeled average # of days to dissemination for the different treatment groups (WT + wAlbB x 2 feeding scenarios). Using the day of 50% of population with dissemination as an estimate of EIP, they then assessed the likelihood of mosquitoes surviving to this critical age point. Interestingly, double fed mosquitoes (WT and wAlbB) were more likely to survive. The difference in survival was more pronounced for WT because of the reduced viral loads due to blocking in wAlbB. In general, the likelihood of surviving EIP was much less for wAlbB.

This is a solid piece of work that shows that blocking (Wolbachia) is generally more powerful than repeat feeding.

Results

The graphics are clear.

Discussion

The authors are self-aware about how the experimental design may have affected their findings.

Suggestions

I find the use of the phrase 'survive EIP' to be awkward. I think you mean survive to EIP or beyond EIP. As written, it sounds like EIP is something to be survived....like an infection.

Response: We thank the reviewer for their comments and thoughtful review. We appreciate the suggestions given and have changed the phrase "survive EIP" to "survive beyond the EIP" or "survive past the EIP" throughout the manuscript.

Reviewer 2

This manuscript by Johnson et al investigates the role of successive bloodmeals on the efficacy of *Wolbachia* in blocking DENV2 transmission. The paper is well written and the methodology is solid. I applaud the authors for investigating the multiple processes that EIP and dissemination feed into. I have mostly minor comments.

Major comment: Given that we've seen success of *Wolbachia* programs like World Mosquito Program, how does this jive with your data? This is worth some Discussion.

Response: We appreciate this comment and have updated the discussion to include more coverage of this important subject (see below with tracked changes). Our data is not contradictory to that of the World Mosquito Program and, in fact, indicates that, at least for *wAlbB*, *Wolbachia*-mediated pathogen-blocking is very effective even under conditions where mosquitoes take multiple blood meals.

“Given this, *Wolbachia* may have an even stronger impact than previously thought when the sequential feeding behavior of *Ae. aegypti* in the wild is taken into consideration. This lends further support to releases of mosquitoes transinfected with *Wolbachia* in the field, and may help explain the reductions in DENV observed in several countries with *Wolbachia* transinfected mosquito releases^{6,28}. Despite this promising data, it is worth noting that our data is primarily from mosquitoes with *wAlbB*, and further study of mosquitoes with different strains of *wMel* is warranted. Additionally, this model does not consider other indirect impacts of *Wolbachia* on mosquito life history traits such as lifespan, fecundity, and feeding frequency^{29–32}.”

Minor comments.

Lines 75-77 – are there a control group of the *Wolbachia* fed group? WT = not transfected? This isn't clear.

Response: We used WT to indicate not transinfected with *Wolbachia*. We have added additional language defining this abbreviation in the abstract, the results, in the discussion, and in figure legends.

Example from Fig 1: “a) Experimental design for initial infection and dissemination studies of single- and double-fed wildtype mosquitoes lacking *Wolbachia* (WT), *wAlbB*, and *wMelM* mosquitoes.”

Does wild type mean from the field, or does it just mean non-transfected? If it is the latter, I think “control” might be a better label to avoid confusion. (I see where it’s defined in the methods, but as those are at the end, “control” might be clearer anyway.)

Response: See comment above. We have added additional language defining this abbreviation in all sections of text and in all applicable figure legends.

Lilnes 81-83: this conclusion is not correct as broadly stated – you did not find a decrease in dissemination, for example, for “*Wolbachia*”, but for one strain. This needs to be clarified throughout to be more specific.

Response: Thank you for pointing this out. We have changed the language here and throughout to be more specific about which groups we saw differences in rather than generalizing to all *Wolbachia*. See below example.

“Our findings show that successive blood feeding results in increased dissemination at 7 days post-infection (dpi) in both the presence and absence of wAlbB *Wolbachia*.”

Line 118-19: your concluding hypothesis here should probably be moved to the Discussion so as not to be misinterpreted as dogma

Response: We appreciate the need to not accidentally inject unproven hypotheses into dogma and have changed the language here to make it clearer that this is only a potential explanation. See quoted section below with tracked changes.

“Thus, increased dissemination rates were associated with increased DENV-2 levels absent large differences in *Wolbachia* density ~~perhaps~~likely due to earlier escape from the midgut, and potentially resulting in more opportunities for replication throughout the mosquito body.”

Lines 151-153: Given the lack of statistical significance, consider softening this conclusion - the next section supports this conclusion better. Also, EIP50 is usually the label given to this (just a consideration for ease of reading). Also, given your low rates in the one group, consider looking at EIPmin – also a good comparison as quicker dissemination even at low levels can affect transmission (see Mayton et al 2020)

We agree with the reviewer that the next section supports the concluding sentence of this paragraph better and have removed this sentence. We have also clarified in the next section that our measure of time until 50% of mosquitoes are disseminated is EIP50.

Example from Fig. 4 legend: “Black lines with arrows mark the timing of 50% dissemination (EIP₅₀) with 95% credible intervals marked by flanking gray lines with arrows.”

Line 207-208 – I think this sentence is incomplete – reflects the interaction of dissemination and mortality?

Response: We have changed this sentence to be clearer. See quoted below with tracked changes.

“This ~~is due to~~reflects the increase in time to dissemination observed in *wAlbB* mosquitoes relative to WT mosquitoes, and reflects the interaction between virus dissemination and mosquito mortality. ~~and demonstrates that changes in dissemination time can interact with mosquito mortality to impact the likelihood of surviving beyond the EIP.~~”

Line 210-212: move to Discussion

We reworded the concluding sentence of this paragraph and moved the previous sentence to the discussion.

“This suggests that *wAlbB* remains effective in inhibiting DENV-2 when considering successive feeding.”

Moved to discussion: “This, ~~suggestsing~~ that *Wolbachia* remains an effective strategy to inhibit DENV-2 transmission even under successive feeding conditions and that. ~~Overall, this suggests that estimates of DENV-2 inhibition by *Wolbachia* based on t~~ traditional single-feeding experiments that fail to account for *Ae. aegypti* successive feeding behavior may underestimate the effectiveness of *wAlbB Wolbachia*.”

Line 410: statistical

Response: Thank you, we have made this change (see below).

“Specific statistical tests are noted in the legend for each graph.”

Reviewer 3

Summary:

The manuscript investigates the relationship between successive bloodmeals and virus dissemination in *Wolbachia*-infected mosquitoes, advancing the state of knowledge from several cited references (e.g., Amuzu et al., 2015, Armstrong et al., 2020; Brackney et al., 2021; Johnson et al., 2023). Specifically, the paper tests the effect of a second non-infectious bloodfeed on the effectiveness of *Wolbachia*-based inhibition of DENV-2 in *Aedes aegypti* mosquitoes in terms of DENV-2 infection and dissemination rates.

A statistical model of the timing of DENV-2 dissemination demonstrates that a second bloodfeed shortens EIP length by 1 day in mosquitoes carrying wAlbB. A second statistical model on the probability of mosquito survival over the EIP to show that a second blood feed increases this probability for both wildtype and wAlbB-infected mosquitoes, but to a much lesser degree for the latter. The authors conclude that models estimating the efficiency of *Wolbachia* blocking based on traditional single-feeding experiments are underestimating *Wolbachia* effectiveness.

Strengths:

The use of statistical modelling to translate classical vector competence measures into epidemiologically meaningful parameters such as EIP and probability of mosquito survival provided valuable insight.

Weaknesses:

- The results reported in the manuscript from Figures 1 to Figure 5A seem to present lower *Wolbachia* effectiveness and therefore worse epidemiological outcomes when double-fed wildtype or wAlbB mosquitoes are compared to their single-fed counterparts. It is surprising that the authors then claim that transmission models that take into account successive bloodfeeding would indicate higher *Wolbachia* effectiveness, which seems to be largely based on Figure 5B. A clearer description of how the odds ratios are calculated, and more explanation behind this rationalisation is needed.

Response: Thank you for this thoughtful summary and review. We first determined the impact of a second blood meal on DENV-2 dissemination in the presence and absence of *Wolbachia*. As the reviewer points out, we compared single-fed vs double-fed for each mosquito colony

separately and generally found higher dissemination rates and shorter EIP in the double-fed groups relative to single-fed groups (**Figs. 1-3**). While these findings indeed show an impact of successive feeding behavior, the more epidemiologically relevant comparison is not within mosquito group, but between mosquito groups (WT vs *wAlbB*). To make these comparisons, we modeled the probability that mosquitoes survive beyond the extrinsic incubation period (**Fig. 4-5**). Given that *Aedes aegypti* mosquitoes take frequent blood meals in the field, we believe that the most relevant comparisons are the odds ratio of *wAlbB* surviving the EIP relative to WT. This is modelled in Fig 5b as a ratio of *wAlbB*:WT surviving the EIP by average lifespan, and we found that the odds ratios in the double-fed scenario are lower as compared to the single-fed scenario. This is due to the relatively stronger increase in dissemination observed in WT mosquitoes after a second blood meal, whereas *wAlbB* is still able to inhibit DENV-2 dissemination (smaller relative increase between SF and DF as compared to WT; **Fig. 5a**). Importantly, this effect is not limited to our estimates of the probability of surviving the EIP, but we would arrive at similar conclusions when using raw data to calculate the odds ratios. We have clarified the rationale for calculating the odds ratios and how odds ratios were calculated.

Edits to the results section discussing Fig 5: “We went on to quantify the epidemiological significance of these factors by calculating the associated odds ratio of surviving past the EIP, associated with *wAlbB* infection status and how that effect was modulated by blood-feeding status (**Fig 5b**). This comparison is particularly relevant as it enables us to determine the odds ratios for surviving the EIP for *wAlbB* relative to WT mosquitoes, under both the traditional single-fed and more natural and epidemiologically relevant double-fed scenarios.”

Edits to methods: “Specifically, we calculated the odds ratio as

$$OR(\text{survive to disseminate}) = \frac{E[\text{Pr}(\text{survive to disseminate})]}{1 - E[\text{Pr}(\text{survive to disseminate})]}.$$

The advantage of an odds ratio for quantifying the effect of *wAlbB* on surviving the EIP is that it accounts for differences in the baseline probability of surviving the EIP for SF and DF mosquitoes.”

• In this study, double-fed wAlbB mosquitoes are always compared to single-fed wAlbB but never to single-fed wildtype mosquitoes. The authors would probably find that double-fed wAlbB vs. single-fed wildtype mosquitoes show reduced *Wolbachia* effectiveness relative to single-fed wAlbB vs. single-fed wildtype mosquitoes. This is also an epidemiologically relevant comparison. The odds ratio calculations, for example, should take this into account too.

Response: We respectfully disagree with this assessment. In earlier experiments we assessed the impact of double-feeding on infection and dissemination (**Figs 1, 3, and 4**). For these experiments, comparing single- to double-fed within group (WT or wAlbB) is the most logical comparison in order to determine the impact of a second blood meal. Later we compared single-fed WT to single-fed wAlbB and double-fed WT to double-fed wAlbB as the more epidemiological relevant comparison to more directly study the impact of *Wolbachia* transinfection on time to transmission aka surviving past the EIP (**Fig 5**). While we expect that the reviewer is correct that when comparing double-fed wAlbB mosquitoes to single-fed WT mosquitoes, double-fed wAlbB will be less effective than single-fed wAlbB at inhibiting *Wolbachia* (due to increased dissemination and higher likelihood for surviving the EIP), we do not believe that this is a relevant comparison given that we do not expect differences in mosquito feeding behavior with *Wolbachia* transinfection. With similar feeding behavior, the most epidemiologically relevant comparison is WT double-feed to wAlbB double-feed, as *Aedes aegypti* mosquitoes feed frequently in the field. This is the comparison that we tested in **Fig. 5b** and we have now clarified the rationale and methods on how the OR for this comparison were calculated (see previous comment). We have added some language clarifying this to the abstract (see below).

“Importantly, the more epidemiologically relevant comparison of the odds of wAlbB mosquitoes surviving beyond the EIP relative to WT, revealed a larger impact of successive feeding on WT than wAlbB.”

• There is not enough novelty nor mechanistic insight in the experimental findings to warrant publication in a broad-interest journal. It should be noted that these findings are not generalisable for all *Wolbachia* strains—the relevance of this manuscript is therefore limited to vector control efforts using the wAlbB. The strong point of the paper is in the transformation of vector biology empirical data into epidemiological significance—this work would be better suited for an epidemiology or vector biology journal. That said, the authors stopped short of performing transmission models themselves to demonstrate that the shortening of EIP truly translates to increased disease incidence.

Response: We believe our work is relevant to a broad audience interested in understanding the factors that may impact the effectiveness of *Wolbachia*. Our work is relevant to individuals working in diverse fields in academia, governments, non-profits organizations, and industry.

differences in timing of dissemination. We have changed the text to provide a little more context for this (see below).

~~“To measure the impact of successive feeding on timing of dissemination, we modelled time to dissemination using To accurately estimate the time at which 50% of WT and wAlbB mosquitoes have disseminated infections, we used empirical data from our time course experiments to model DENV-2 dissemination (Fig 4). To quantitatively assess the differences between groups, we then estimated the time at which 50% of WT and wAlbB mosquitoes had disseminated infection (EIP₅₀; Fig 4).”~~

- Line 242-245: That *Wolbachia* density was not affected by number of blood feeding has also been reported in Amuzu et al (2015). This would be an appropriate place to cite that work for this finding.

Response: Thank you for pointing this out. We have added this additional reference and reworded this section slightly (see below).

“In agreement with a previous study we found little variation in *wAlbB* or *wMeIM* *Wolbachia* levels by feeding or infection or dissemination status (**Fig 2b, 2d, S1a-d, S2b, S3a, S3b**)¹⁹. Thus, we propose that the differences in virus titer we observed are not correlated with *Wolbachia* density, as has previously been suggested²⁷.”

- Line 462-465: How is OR ratio calculated? Please rephrase for clarity.

Response: We have added additional language clarifying how the odds ratio was calculated (see below).

“Specifically, we calculated the odds ratio as

$$OR(\text{survive to disseminate}) = \frac{E[\text{Pr}(\text{survive to disseminate})]}{1 - E[\text{Pr}(\text{survive to disseminate})]}.$$

The advantage of an odds ratio for quantifying the effect of *wAlbB* on surviving the EIP is that it accounts for differences in the baseline probability of surviving the EIP for SF and DF mosquitoes.”

- Figure 4c-d: Please, either through annotations on the graphs or in the figure legend, state the values of days post infections where the grey and black arrows fall.

Response: We added these numbers to the figure legend (see below) and in doing so noticed one area in the result section text where figure 4 graphs were referenced in the wrong order. We fixed this as well.

“Black lines with arrows mark the timing of 50% dissemination with 95% credible intervals marked by flanking gray lines with arrows. For c) 50% dissemination = 8.38 days post-infection (95% credible interval [CrI]: 7.72-9.01. For d) 50% dissemination = 6.86 days post-infection (95% credible interval [CrI]: 6.03-7.62. Blue = single-fed, red = double-fed. SF = single-fed, DF = double-fed.”

- The number of replicates are only indicated in table S2. Please, add the information to the figure legend, including the number of mosquitoes analysed in each replicate.

Response: We have modified the legend of each figure to include this information (see below example from **Fig 2** with tracked changes). In making these additions we noticed an error with figure 2b where some datapoints from the wMelM group were inadvertently excluded. These datapoints did not alter our conclusions but we have corrected this error and replaced the graph with a new version including these datapoints.

“Figure 2: Comparison of DENV-2 titers and relative *Wolbachia* densities at 7 dpi by feeding and dissemination status. a) Body DENV-2 titers in single- and double-fed wildtype mosquitoes lacking *Wolbachia* (WT), wAlbB and wMelM mosquitoes. WT SF n = 34, WT DF n = 42, wAlbB SF n = 46, wAlbB DF n = 35, wMelM SF n = 11, wMelM DF n = 27. b) Relative body *Wolbachia* density in single- and double-fed wAlbB and wMelM mosquitoes. Numbers tested are as follows; wAlbB SF n = 74, wAlbB DF n = 61, wMelM SF n = 119, and wMelM DF n = 148. c) Body DENV-2 titers in wAlbB mosquitoes by dissemination status and feeding status. Total Dis n = 46 and Not Dis n = 35. Dis SF n = 18, Dis DF n = 28, Not Dis SF n = 28, and Not Dis DF n = 7. d) Relative body *Wolbachia* densities in wAlbB mosquitoes by infection, dissemination, and feeding status. Total Dis n = 46, Not Dis n = 35, and Not Inf n = 54. Dis SF n = 18, Dis DF n = 28, Not Dis SF n = 28, Not Dis DF n = 7, Not Inf SF n = 28, and Not Inf DF n = 26. Comparisons were made using Mann-Whitney U tests (a, b, and c) or a Kruskal-Wallis test with Dunn’s multiple comparisons (d). * = $p \leq 0.05$, ** = $p \leq 0.01$, *** = $p \leq 0.001$, **** = $p < 0.0001$. Blue = single-fed, red = double-fed. GE = genome equivalents. Dis = mosquitoes with a disseminated infection (i.e., dengue infected bodies and legs), Not Dis = mosquitoes with a non-disseminated infection (i.e., dengue infected bodies, but not legs), Not Inf = mosquitoes that were exposed but are not infected with DENV-2. Lines indicate median with 95% confidence interval. Data was collected across 4 replicates for WT and wAlbB groups and 5 replicates for wMelM groups.”

- Check for consistency in use of wMel and wMelM in text and figures

Response: Thank you for catching this. We have changed wMel to wMelM in Table S2 and Figure 1A. In making these changes we also found some instances where we used wAlb instead of wAlbB. These were corrected in Table S1, Figure 4c and 4d, and Figure 5a and 5b.

- Line 27: Correct to “Wolbachia transinfected mosquitoes”

Response: Thank you for pointing this out. We have corrected this line to include “mosquitoes” (see below).

“We found that both WT (mosquitoes without Wolbachia) and *Wolbachia* transinfected mosquitoes had increased DENV-2 dissemination 7 days post-infection as well as higher body titers of DENV-2 in the double-fed groups.”

- Line 199: The logical order of this sentence is inverted. WT mosquitoes are more likely to survive the WIP than wAlbB mosquitoes because wAlbB inhibits DENV-2 infection and dissemination. It would be more accurate to say “indicating that wAlbB reduces the likelihood of DENV-2 transmission”.

Response: We have made this change to clarify our phrasing (see below).

“WT mosquitoes were more likely to survive beyond the EIP than wAlbB mosquitoes, regardless of number of blood meals, indicating that wAlbB inhibits DENV-2 infection and dissemination reduces the likelihood of DENV-2 transmission dissemination (Fig 5a).”

- Line 388: Indicate the gene used to quantify Wolbachia by qPCR.

Response: Thank you for pointing this out. We have added this information. See below quote.

“Samples were screened for *Wolbachia* genome equivalents using primers targeting a DNA-directed RNA polymerase subunit beta/betagene with locus tag WD_RS06155 and compared to *Ae. aegypti* S6 DNA copies using the NEB Luna Universal qPCR kit on the CFX Connect Real-Time PCR Detection System and protocols and primers were modified from Lau et. al and Lee et. al (Table S3)^{37,38}.”